# Effect of Indole-3-Acetic Acid on Tomato Plant Growth

**DOI:** 10.3390/microorganisms10112212

**Published:** 2022-11-09

**Authors:** Laiana Lana Bentes Lobo, Maura Santos Reis de Andrade da Silva, Tereza Cristina Luque Castellane, Rogério Falleiros Carvalho, Everlon Cid Rigobelo

**Affiliations:** 1Agricultural and Livestock Microbiology Graduation Program, School of Agricultural and Veterinary Sciences, São Paulo State University (UNESP), São Paulo 01049-010, Brazil; 2School of Agricultural and Veterinary Sciences, São Paulo State University (UNESP), Rod. Prof. Paulo Donato Castellane km 5, Jaboticabal 14884-900, Brazil; 3Department of Applied Biology to Agriculture, Faculty of Agrarian and Veterinary Sciences, Access Route Prof. Paulo Donato Castellane, Vila Industrial, Jaboticabal 14884-900, Brazil

**Keywords:** *Bacillus subtilis*, *Azospirillum brasilense*, tomato

## Abstract

Plant growth-promoting bacteria have several abilities to promote plant growth and development. One of these skills is the synthesis of indole-3-acetic acid (IAA), which mainly promotes root and shoot development. The bacteria *Bacillus subtilis* and *Azospirillum brasilense* have been widely used in agriculture with this function. However, little is known about whether the joint inoculation of these bacteria can reduce plant development by the excess of IAA produced as a result of the joint inoculation. The objective of the present study was to verify the effect of IAA on the inoculation of *B. subtilis* and *A. brasilense* in three tomato genotypes. The Micro-Tom genotype without mutation for IAA synthesis, *Entire*, has high sensitivity to IAA, and the *diageotropic* genotype (*dgt*) has low sensitivity to IAA. The results show that the plant parameter most sensitive to microbial inoculation is the number of roots. No treatment increased the shoot dry mass parameters for the Micro-Tom genotype and *dgt*, root dry mass for the Micro-Tom genotype, plant height for the Micro-Tom and *Entire* genotypes, root area and root volume for the genotype *dgt*. The Azm treatment reduced plant height compared to the control in the *dgt*, the BS + Azw and BS + Azm treatments in the Micro-Tom genotype and the Azw + Azm treatment in the *dgt* genotype reduced the plant diameter compared to the control. BS and BS + Azw reduced the number of roots in the Micro-Tom. The results strongly support that the mixture of *B. subtilis* and *A. brasilense* can reduce some parameters of plant development; however, this effect is possibly an interference in the mode of action of growth promotion of each isolate and is not related to an excess of IAA produced by the bacteria.

## 1. Introduction

Plant growth-promoting bacteria (PGPB) have several abilities that promote plant growth and development. These abilities may be the synthesis of phytohormones such as indole-3-acetic acid (IAA) [1], phosphorus solubilization [2], nitrogen fixation [3], production of siderophores [4] and induction of systemic resistance [5].

The bacteria capable of producing IAA belong to numerous genera Aeromonas, Azospirillum, Enterobacter, Erwinia, Pantoea, Pseudomonas, Flavobacterium, Flavobacteriumbacterium, Microbacterium, Bacillus, Bradyrhizobium, Streptomyces, Azotobacter, Klebsiella, Alcaligenes, Brevebacteria, Rhodocystia, Rhodocyst and Dickeya. These bacteria live as free-living microorganisms in the soil or as endophytes, colonizing plant tissue [1,6].

Most endophytic bacteria are capable of producing IAA. This phytohormone plays an important role as a molecule of interaction between the plant and the microorganism, in addition to improving the survival of bacteria [7,8]. IAA increases the survival rate of bacteria in situations of environmental stress, such as excessive ultraviolet rays, salinity, acidity, and high and low temperatures. IAA also plays an important role in the chemotaxis of bacteria, a movement used for the bacteria to reach the plant roots [9,10].

PGPB can be inoculated into agricultural crops individually, or as a mixture of microorganisms. Both forms bring benefits and losses. Sometimes the failure in the establishment of microorganisms with plant growth promotion abilities occurs due to the noninteraction between the microorganism and the host plant due to the lack of connection between the cell membrane receptors of the microorganisms and the host cells [11,12]. In this sense, products containing more than one microorganism in their formulations would increase the chance of at least one microorganism of the formulation interacting with the host plant. On the other hand, more studies are needed to answer this question. Is it possible that products based on mixtures of various microorganisms, many of them phytohormone producers, do not hinder the growth and development of plants due to the excess of indole-3-acetic acid? [13].

IAA is a phytohormone and, like all phytohormones, is an organic molecule that, depending on the concentration, promotes or inhibits plant growth [14,15]. Some studies have shown the benefits and effects of promoting plant growth in the use of biological products based on mixtures of various microorganisms [16,17]. Other studies show that these mixtures of microorganisms can cause no effect or even damage in plant growth [13,18].

## 2. Objective

The objective of the present study was to verify whether inoculation with a mixture of *B. subtilis* and *A. brasilense*, both indole-3-acetic acid-synthesizing bacteria, impairs the growth and development of tomato plants of three different genotypes.

## 3. Materials and Methods

### 3.1. Bacterial Isolates

The following bacterial isolates were used: *Bacillus subtilis* (GenBank deposit number) MZ133755; *Azospirillum brasilense* wild-MZ133758; *Azospirillum brasilense* mutant-MZ133759, belonging to the collection of the Laboratory of Soil Microbiology, FCAV-UNESP, campus of Jaboticabal. For experimental use, bacteria were routinely streaked on plates containing NA (nutrient agar) culture medium.

### 3.2. Tomato Genotypes

Tomato seeds (*Solanum lycopersicum* L.) with the Micro-Tom variety and its *dgt* (low sensitivity to *auxin/defective* gene for cyclophilin biosynthesis and genotype *Entire* (with high auxin biosynthesis) mutant belong to the Laboratory of Plant Physiology, Department of Biology Applied to Agriculture at FCAV-UNESP, campus of Jaboticabal. Seeds of MT, *dgt* and *Entire* genotypes were multiplied to maintain the laboratory genotype bank.

In the *Entire* tomato genotype (*Solanum lycopersicum* L), locus (e) controls leaf morphology, and the dominant (e) recessive allele and allele of the locus produce pinnate compounds and complex reduced leaves. The leaf morphology of these transgenic lines was similar to the leaf morphology of the *Entire* tomato mutant [19].

In the *diageotropic* genotype (*dgt*), the mutant gene was *Solyc 01g111170*, and plants with the mutant allele had low sensitivity to auxin. The tomato *dgt* genotype has hyponastic leaves and plagiotropic roots with fewer branches and slender stems [20].

### 3.3. Greenhouse Experiment

New seeds of each tomato genotype were produced to be used in the experiment. For this purpose, in the greenhouse, seeds from the laboratory collection were germinated in trays containing a 1:1 mixture commercial substrate (Plantmax HT) Eucatex producer -Brazil with expanded vermiculite supplemented with 1 g L^−1^; 10:10 NPK and 4 g L^−1^ commercial dolomitic limestone (Ca + Mg). After 15 days, under the same substrate conditions, plants were transferred to pots with a capacity of 1 L, remaining for 90 days to collect fruits and seeds used in the experiment. After that, the new seeds were used in the experiment as described above. Table 1 shows the description of the treatments in which the control treatment (without inoculation) consisted of *B. subtilis* (BS), *A. brasilense* wild (Azw) without transformation, *A. brasilense* mutant (Azm) and the mixtures *B. subtilis* + *A. brasilense* wild (BS + Azw), *B. subtilis* + mutant *A. brasilense* (BS + Azm), *B. subtilis* + exogenous auxin 400 µg mL^−1^ (BS + IAA Azw), *A. brasilense* + exogenous auxin (30 µg mL^−1^) (Azw + IAA BS) and *A. brasilense* wild + *A. brasilense* mutant (Azw + Azm).

### 3.4. Exogenous Auxin

The bacteria *B. subtilis*, wild-type *A. brasilense* and mutant *A. brasilense* were grown in culture medium for 24 h at 28 °C. After bacterial growth, the media were filtered and centrifuged. In the centrifuged medium and in the supernatant, the concentrations of IAA were measured with the aid of HPLC. In the *B. subtilis* + IAA Azw mixture, Azw was replaced by the amount of IAA found in the extract extracted from the growth medium of this bacterium, approximately 400 µg mL^−1^. The same was done in the Azw + IAA BS treatment, replacing the corresponding extract with 30 µg mL^−1^ IAA for *B. subtilis*. These treatments were performed to measure the effect of IAA-producing bacteria and IAA alone.

### 3.5. In Vitro Auxin Biosynthesis

The in vitro indole-3-acetic acid production analysis was performed using HPLC. Each bacterial colony and mixture were applied to 20 mL of dextrose yeast glucose sucrose (DYGS) medium containing (in g L^−1^): glucose, 2; peptone, 1.5; yeast extract, 2; potassium dihydrogen phosphate, 0.5; magnesium sulfate heptahydrate, 0.5 [21], supplemented with 5 mM L-tryptophan and incubated for 48 h at 28 °C under constant stirring at 120 rpm in the absence of light. After growth, 5 mL of each culture was centrifuged at 10,000 rpm for 10 min. The samples (15 mL) were collected for centrifugation (Sorvall centrifuge at 16.266× *g* for 30 min at 4 °C) and then concentrated under vacuum using a centrifugal evaporator to a volume of 1 mL. Then, they were filtered through a cellulose ester filter (Millipore Corp., Burlington, MA, USA) with a pore size of 0.45 μm and injected in triplicate into an HPLC equipped with an RID detector (Shimadzu RID model 10A). The injected samples were eluted in 20 μL with a mobile phase of acetonitrile:water (75:25 *v*:*v*) under the following chromatographic conditions: 35 °C injection temperature, 20 μL injection volume, 1.0 mL/min flow. The culture medium at time zero was used as a negative control. Four concentrations (50, 25, 12.5, 6.25 μmol/L) were prepared for each indole compound standard indole-acetic acid, indole-pyruvic acid, indole-lactic acid, and indole-acetaldehyde for quantification and then analyzed by HPLC. A Shimadzu Class-VP chromatography data system was used for data acquisition and data analysis. Means and standard deviations were calculated, and SigmaPlot 11.0 was used to generate graphs.

### 3.6. Inoculations in Seeds and Plants

For inoculations, each bacterial isolate was cultured in a 125 mL Erlenmeyer flask containing 50 mL of nutrient broth (meat extract: 1.0 g L^−1^, yeast extract: 2.0 g L^−1^, peptone: 5.0 g L^−1^, NaCl: 5.0 g L^−1^, pH: 6.8 ± 0.2), and cultures were incubated at 28 °C for 24 h for standardized growth of 10^8^ colony-forming units (CFU) per mL. The seeds were sterilized prior to inoculation. The seeds were washed with tapwater and deionized water and dried on absorbent towels. Then, 2 g of seeds were transferred aseptically to a sterile beaker, washed two times with sterile distilled water and sterilized using 0.2 g% HgCl2 for 30 s. Then, the seeds were washed six times with distilled water [22]. After sterilization, seeds were immersed in bacterial suspensions and stirred at 180 rpm and 28 °C for approximately 30 min. Control treatments consisted of seeds immersed only in the culture medium, free from bacteria, under the same conditions. Throughout the experiment, four inoculations were carried out using 1 mL of each bacterial inoculum according to the treatments in the plants. The period between the inoculations was 15 days.

### 3.7. Azospirillum brasilense Mutant

*A. brasilense* and mutant *A. brasilense* (IAA knockout) for auxin/IAA gene strains were used, which belong to the collection of microorganisms of the Laboratory of Soil Microbiology (LSM)—FCAV/UNESP, campus of Jaboticabal. Both *A. brasilense* strains were gently transformed by Dr. Carl Bauer from the Indian University in the United States.

The mutation method used for the transformation of mutant *A. brasilense* was conjugation, using the *E. coli* HB101 strain to donate the suicidal plasmid pGS9 [23] carrying the Tn5 transposon kanamycin resistance genes for Sp. *F94*. Conjugation was performed using the filter membrane method, and the transformed isolates were selected on Luria Bertani agar supplemented with kanamycin and rifamycin both at a concentration of 40 µg mL^−1^.

### 3.8. Experimental Design

The experiment was carried out in a completely randomized design with nine treatments for each tomato genotype and six replicates (Table 1).

The seeds were planted in each pot according to the treatment. The duration of the experiment was 60 days.

### 3.9. Plant Height and Diameter

After 60 days, the plants of each pot were collected, the plant height was measured with a ruler, and the plant diameter was measured with the aid of a caliper.

### 3.10. Root Length, Area and Density

The roots were separated from the shoots, washed, and kept in a test tube containing alcohol (30%). The roots were spread in a layer of water in a transparent tray (30 cm × 20 cm), and images were captured at 400 dpi with a professional Epson Expression 11000XL scanner system. The images were analyzed for root length, root area, root volume and number of roots using WinRHIZO ^TM^ Arabidopsis software (Reagent Instruments Inc., Quebec City, QC, Canada).

### 3.11. Shoot and Root Dry Mass

To obtain dry mass, the material was kept in a paper bag and placed in a forced air oven at 55 °C for 72 h. Subsequently, the root dry mass and shoot dry mass were obtained using an analytical scale (Denver Instrument Company AA-200, Bohemia, NY, USA) with an accuracy of 0.0001 g.

## 4. Results

The amounts of indole compounds measured in the supernatant and in the concentrated extract by HPLC were 76.34 µg IAA mL^−1^ and 416.41 µg IAA mL^−1^ for Azw, 8.41 µg IAA mL^−1^ and 140.89 µg IAA mL^−1^ for Azm, and 4.86 and 30.56 µg IAA mL^−1^ for Azw, respectively.

### 4.1. Micro-Tom Genotype

For the Micro-Tom tomato genotype, there was no significant difference (*p* > 0.05) in the shoot dry mass parameter between treatments (Figure 1a). There was also no difference between the treatments and the control (*p* > 0.05) for the parameters plant height, root dry mass and root volume (Figure 1b,d,g). For the plant diameter, the lowest values were found in the BS + Azm and BS + AZm treatments (*p* < 0.05) compared to the control, and there was no difference between the other treatments (*p* > 0.05) (Figure 1c). For the root length parameter, the highest values were found in the Azm and Azw + IAA BS treatments (*p* < 0.05), and the other treatments did not differ from the control (*p* > 0.05) (Figure 1e). For the root area parameter, the highest value was found for the Azm treatment (*p* < 0.05). All other treatments did not differ from the control (*p* > 0.05) (Figure 1f). For root volume, there was no significant difference between the treatments and the control (Figure 1g). For the parameter number of roots, the BS treatment had the lowest value compared to the control, and the only treatment that showed the highest value compared to the control and the other treatments was Azw + IAA BS (Figure 1h).

In the principal component analysis, the Azw + IAA BS treatments had a positive correlation with the parameters root number, root length, root area and root volume. The Azw, BS + IAA Azw, BS + IAA Azw, BS + Azw, and Azw + Azm treatments had a positive correlation with the parameters plant height and root diameter (Figure 2).

The analysis of the hierarchical component shows the formation of eight groups, with the treatments Azw, BS + IAA Azw, Azw + Azm and BS + Azw being the closest and BS + Azm being the most distinct (Figure 3).

### 4.2. Entire Genotype

For the *Entire* tomato *genotype* for the shoot dry mass parameter, the highest values were found for the Azw + IAA BS and Azw + Azm treatments (*p* < 0.05). The other treatments did not differ from the control or from each other (*p* > 0.05) (Figure 4a). For the plant height parameter, there was no significant difference between the treatments and the control (Figure 4b). For the plant diameter, the highest values were found in the Azw, BS + Azw, Azw + IAA BS and Azw + Azm treatments (*p* < 0.05). The other treatments did not differ from the control. (Figure 4c). For the root dry weight parameter, the highest and only value that differed from the control and from the other treatments was Azw + IAA BS (*p* < 0.05). The other treatments did not differ from each other (*p* > 0.05) (Figure 4d). For the root length parameter, the highest values were found in the BS + IAA Azw, Azw + IAA BS and Azw + Azm treatments (*p* < 0.05). There was no significant difference between the other treatments (*p* > 0.05) (Figure 4e). For the root area parameter, the highest values were found in the BS + IAA Azw, Azw + IAA BS and Azw + Azm treatments (Figure 4f). For the root volume parameter, the highest values were found in the Azw + IAA BS and Azw + Azm treatments. The other treatments did not differ from each other (*p* > 0.05) (Figure 4g). For the parameter root number, the treatments that did not differ from the control were only BS and BS +Azm (*p* < 0.05), and the other treatments showed higher values than the control (*p* < 0.05) (Figure 4h).

In the principal component analysis, the Azw + IAA, Azw and BS + Azw treatments had a positive correlation with the parameters plant height, root dry mass, plant diameter and shoot dry mass. The BS + IAA, Azw, and Azw + Azm treatments had a positive correlation with the parameters root volume, root area, root length and number of roots (Figure 5).

In the hierarchical clustering analysis, the Azw + IAA BS and Azw + Azm treatments were the most distinct compared to the other treatments (Figure 6).

### 4.3. Genotype dgt

There was no difference in the *diageotropic* genotype (*dgt*) for the shoot dry mass parameter (*p* > 0.05) between the control treatment and the other treatments (Figure 7a). For the plant height parameter, the highest value was found in the BS + IAA Azw treatment (*p* < 0.05), and the lowest value and only treatment that differed from the control was Azm (*p* < 0.05). There was no significant difference (*p* > 0.05) between the other treatments (Figure 7b). For the parameter plant diameter, the lowest value and the only treatment that differed from the control was Azw + Azm (Figure 7c). For the root dry mass parameter, the highest value and the only parameter that differed from the control was BS + IAA Azw (*p* < 0.05). There was no difference between the other treatments (Figure 7d). For the root length parameter, the highest values were found in the Azw and BS + IAA Azw treatments compared to the control (*p* < 0.05). There was no significant difference (*p* > 0.05) between the other treatments (Figure 7e). For the root area parameter, there was no difference between the treatments and the control (Figure 7f). For the root volume parameter, there was no difference between the control and the other treatments (*p* < 0.05) (Figure 7g). For the parameter number of roots, the highest values were found in the Azw and Azw + Azm treatments. There was no difference between the other treatments and the control (Figure 7h).

The principal component analysis shows that the Azw treatment had a positive correlation with the parameters number of roots, root length, and root area and that the Azw + IAA BS and BS + IAA Azw treatments had a positive correlation with the parameter plant height.), root dry mass, root volume, shoot dry mass and plant diameter (Figure 8).

In the hierarchical cluster analysis, the closest groups were Azm and Azw + Azm. The next groups were BS + Azm, BS + Azw, Azw + IAA BS, BS and control. The groups most distinct from the others were Azw and BS + IAA Azw (Figure 9).

## 5. Discussion

The inoculations with the bacteria *B. subtilis*, *A. brasilense* wild and *A. brasilense* mutant promoted increases in most of the measured plant parameters compared to the control treatment that did not receive inoculation. However, it is important to note that the inoculation of PGPB synthesizing IAA did not differ from the control treatment for the parameters shoot dry mass in the Micro-Tom and *dgt* genotypes, root dry mass for the Micro-Tom genotype, plant height for the genotypes Micro-Tom and *Entire*, root area for the *dgt* genotype and root volume for the Micro-Tom and *dgt* genotypes. The inoculation of the mixtures of these bacteria also promoted the reduction in the development of some parameters, such as the reduction in plant diameter in the BS +Azw and BS +Azm treatments for Micro-Tom and Azw +Azm in the *dgt* genotype, and there was a reduction in the height of the inoculated plant with Azm and BS reduced the number of roots in the Micro-Tom genotype (Table 2).

The main effect of inoculation of IAA-producing bacteria in plants is usually an increase in root development and, as a consequence, an increase in the efficiency of obtaining water, nutrients, and soil volume exploitation, thus allowing a reduction in the dose of fertilizers [24,25]. However, some studies have shown that inoculation of a mixture of IAA-producing microbial isolates may also not effectively respond to the replacement of an agricultural input because it does not promote or even decrease plant development. Felici [18] used a mixture of *B. subtilis* and *A. brasilense* bacteria in tomato plants and found that the combination of the two bacteria did not improve plant growth compared to inoculation of the bacteria separately. These authors support the hypothesis that these two bacteria may operate differently in the modulation of root growth. In seedlings inoculated with the mixture of *B. subtilis* and *A. brasilense*, a strong inhibition of primary root elongation was observed. This inhibitory effect also induced a process of resistance to morphogenesis [26,27]. Therefore, [18] concluded that the presence of the two microorganisms can directly or indirectly alter the internal hormonal content of the root, interfering with its normal morphogenesis. Similar results were found by [13]. These studies suggest that the reduction in some plant parameters promoted by the mixture of these bacteria is not related to the excess of IAA produced by the bacteria but by an interference in the mode of action of the isolates.

The synthesis of IAA is not only used for plant growth and development, but low concentrations of IAA produced by bacteria can interfere with the response of certain microorganisms; for example, they can regulate the formation of semi mycelial structures and antimicrobial activity in some strains of bacterium *Streptomyces*. Thus, the secretion of IAA in the rhizosphere produced by bacteria and plants is a signal for *Streptomyces* spp. to improve their antimicrobial production activity, improving their efficiency in establishing their colonization niche against other microorganisms [28,29].

The results of the present study reinforce the suggestions of previous studies and show that although the mixture of bacteria has reduced the plant development of some parameters, this effect is not related to the synthesis of IAA. For example, the BS + Azw and BS + Azm mixtures reduced the plant diameter in the Micro-Tom genotype, and the BS + Azm mixture produced almost three times less IAA than BS + Azw (Figure 1e). On the other hand, for the root dry mass parameter, there was no difference for the same treatments BS + Azw (30.56 µg IAA mL^−1^ + 416.41 µg IAA mL^−1^) and BS + Azm (30.56 µg IAA mL^−1^ + 140.89 µg IAA mL^−1^), even though the latter produced less IAA (Figure 1d). These results show that the concentration of IAA is not the only determining factor but that each plant parameter responds differently to inoculations. Another interesting result occurred in the *Entire* genotype (which is sensitive to IAA) for the shoot dry mass parameter. There was no difference between the BS + Azw treatments that would theoretically produce (30.56 µg IAA mL^−1^ + 416.41 µg IAA mL^−1^) or BS + Azm (30.56 µg IAA mL^−1^ + 140.89 µg IAA mL^−1^), and there was a difference between BS + IAA Azw (30.56 µg IAA mL^−1^ + 400.00 µg IAA mL^−1^) and Azw + IAA BS (416.41 µg IAA mL^−1^ + 30 µg IAA mL^−1)^ (Figure 4a). These treatments have the ability to produce the same amount of IAA in the plant or less, and even so, there was no difference in a sensitive genotype, which is the *Entire* genotype. The application of plant growth-promoting microorganisms, including IAA-producing bacteria, is an excellent alternative to address the challenges of global agriculture due to the possibility of reducing production costs, doses of mineral fertilizers and environmental impacts [30,31]. However, in some situations, these microorganisms may fail for this purpose. This failure occurs due to the failure of the plant growth-promoting microorganism to colonize the rhizosphere and subsequently the plant tissues to express the effect growth promoter. In this sense, the use of formulations containing more than one microorganism would increase the chances of successful colonization [12]. However, it is necessary to better understand whether these mixtures or combinations of different IAA-producing microorganisms do not reduce plant development due to excess IAA. The results of the present study show that this reduction may occur but is not related to IAA but is most likely the interference of each microorganism in the mode of action and in the growth promotion effect.

## 6. Conclusions

The inoculation of the mixture of *B. subtilis* and *A. brasilense* bacteria can reduce some plant parameters, such as plant height and stem diameter and the number of roots; however, this negative effect is not related to the excess of IAA produced by the bacteria but is probably an interference of each bacterium in the mode of interaction and in the growth promotion effect with the plant.

## Figures and Tables

**Figure 1 microorganisms-10-02212-f001:**
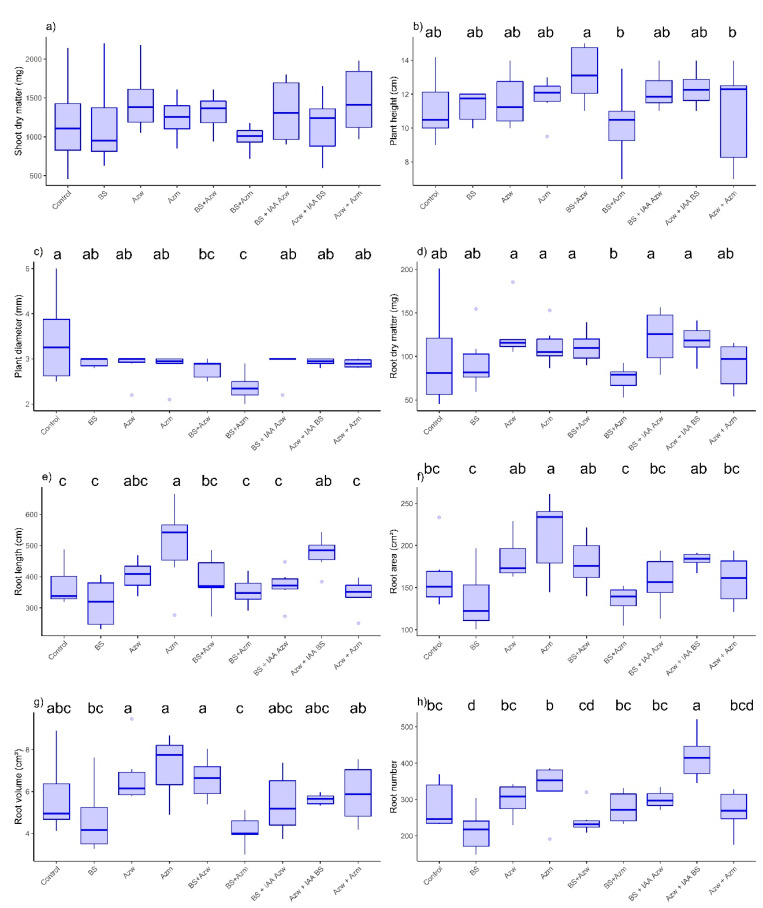
Boxplots for growth evaluation of Micro-Tom tomato plants inoculated with auxin-synthesizing bacteria. (**a**) shoot dry mass, (**b**) plant height, (**c**) plant diameter, (**d**) root dry matter, (**e**) root length, (**f**) root area, (**g**) root volume, (**h**) number of roots. Means followed by the same lowercase letter or absence of letters do not differ by Tukey’s test (*p* < 0.05). Control, without inoculation; BS, *B. subtilis;* Azw, *A. brasilense* wild; Azm, *A. brasilense* mutant; BS + Azw, *B. subtilis* + *A. brasilense* wild; BS + Azm, *B. subtilis* + *A. brasilense* mutant; BS + IAA Azw, *B. subtilis* + exogenous auxin (400 µg mL^−1^); Azw + IAA BS, *A. brasilense +* exogenous auxin (30 µg mL^−1^); Azw + Azm, *A. brasilense* wild *+ A. brasilense* mutant.

**Figure 2 microorganisms-10-02212-f002:**
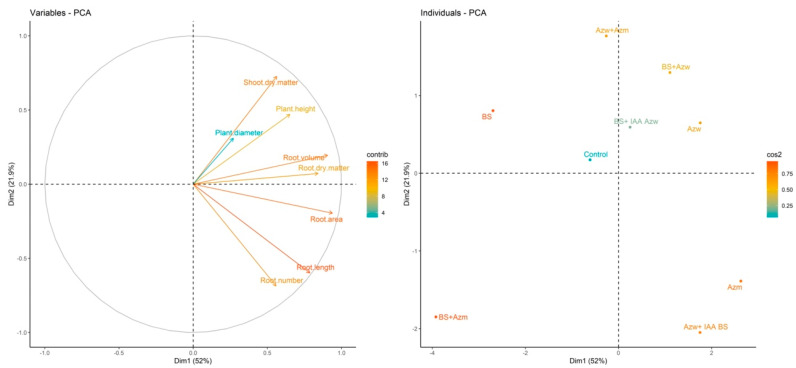
Main component analysis (PCA) for variables and treatments (individuals). Values on Axes 1 and 2 represent the percentage of total variance explained by axes for the Micro-Tom genotype.

**Figure 3 microorganisms-10-02212-f003:**
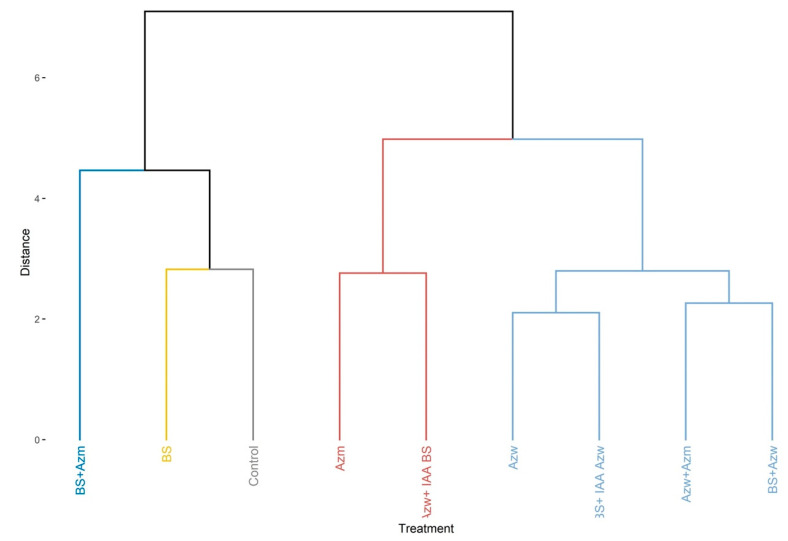
Hierarchical clustering of principal components (HCPCs) of inoculations tested.

**Figure 4 microorganisms-10-02212-f004:**
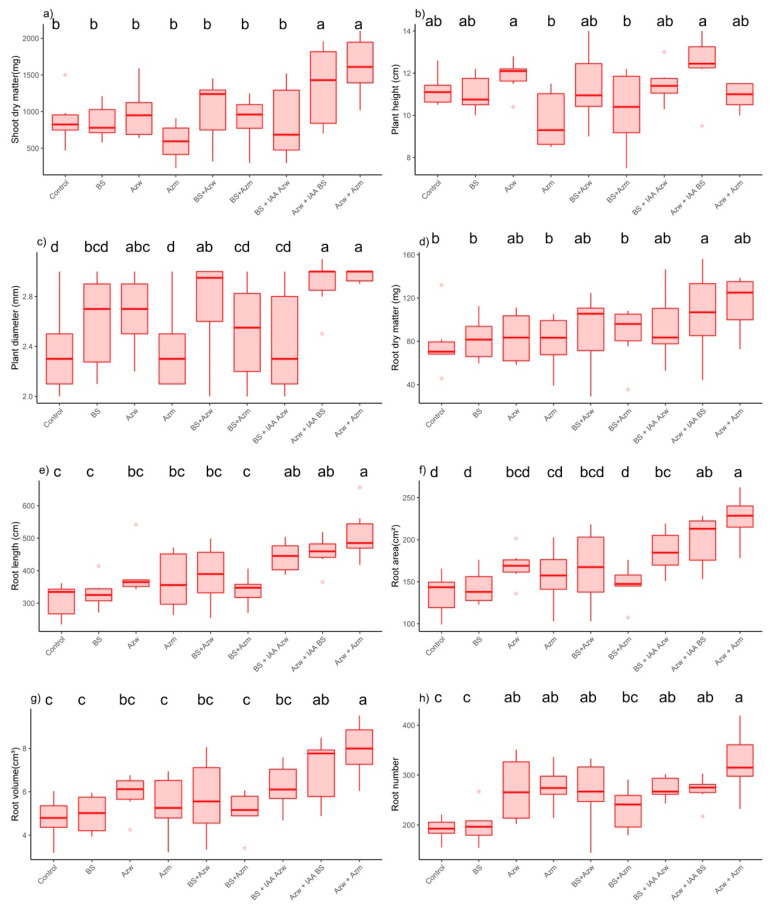
Boxplots of the *Entire* tomato plant growth evaluations inoculated with auxin-synthesizing bacteria. (**a**) shoot dry mass, (**b**) plant height, (**c**) plant diameter, (**d**) root dry matter, (**e**) root length, (**f**) root area, (**g**) root volume, (**h**) number of roots. Means followed by the same lowercase letter or absence of letters do not differ by Tukey’s test (*p* < 0.05). Control, without inoculation; BS, *B. subtilis;* Azw, *A. brasilense* wild; Azm, *A. brasilense* mutant; BS + Azw, *B. subtilis* + *A. brasilense* wild; BS + Azm, *B. subtilis* + *A. brasilense* mutant; BS + IAA Azw, *B. subtilis* + exogenous auxin (400 µg mL^−1^); Azw + IAA BS, *A. brasilense +* exogenous auxin (30 µg mL^−1^); Azw + Azm, *A. brasilense* wild *+ A. brasilense* mutant.

**Figure 5 microorganisms-10-02212-f005:**
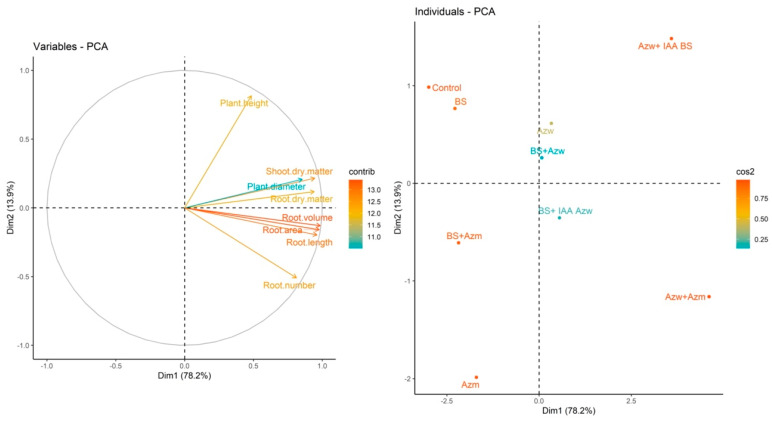
Main component analysis (PCA) for variables and treatments (individuals). Values on axes 1 and 2 represent the percentage of total variance explained by axes for the *Entire genotype*.

**Figure 6 microorganisms-10-02212-f006:**
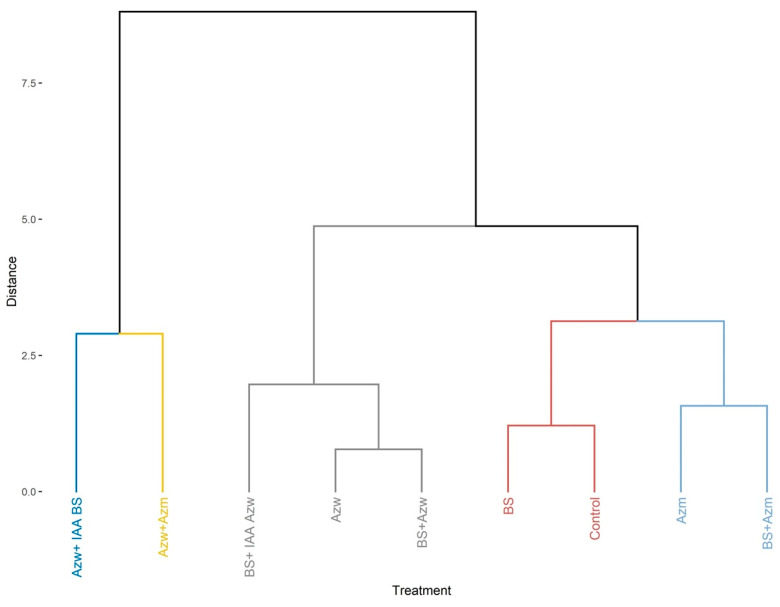
Hierarchical clustering of principal components (*HCPCs*) of inoculations tested for the *Entire* genotype.

**Figure 7 microorganisms-10-02212-f007:**
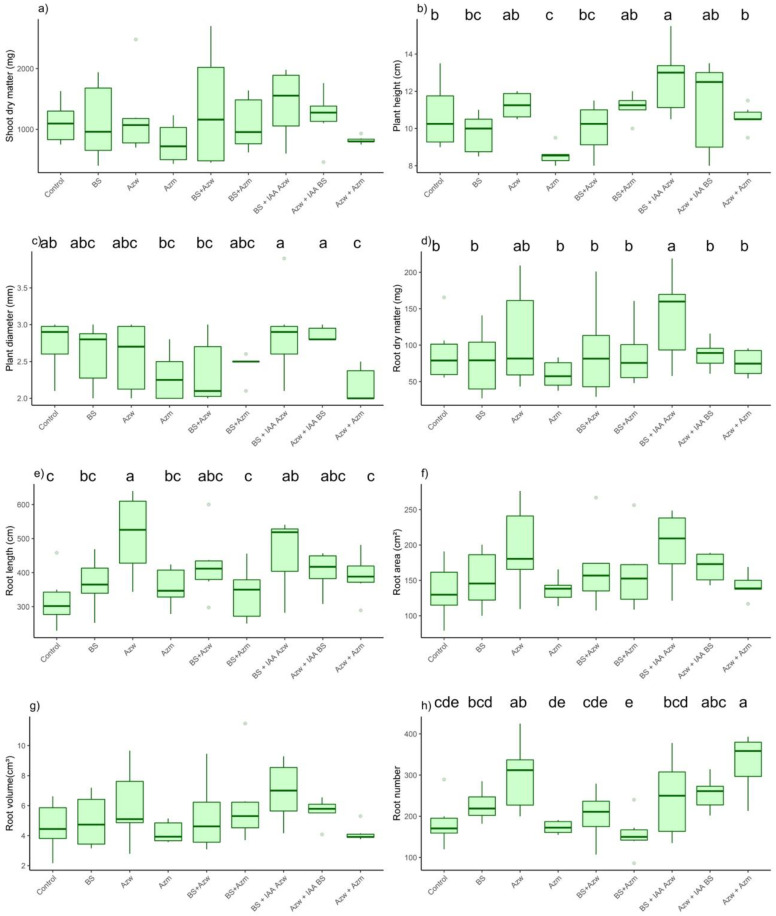
Boxplots for growth evaluation of *dgt* tomato plants inoculated with auxin-synthesizing bacteria. (**a**) shoot dry mass, (**b**) plant height, (**c**) plant diameter, (**d**) root dry matter, (**e**) root length, (**f**) root area, (**g**) root volume, (**h**) number of roots. Means followed by the same lowercase letter or absence of letters do not differ by Tukey’s test (*p* < 0.05). Control, without inoculation; BS, *B. subtilis;* Azw, *A. brasilense* wild; Azm, *A. brasilense* mutant; BS + Azw, *B. subtilis* + *A. brasilense* wild; BS + Azm, *B. subtilis* + *A. brasilense* mutant; BS + IAA Azw, *B. subtilis* + exogenous auxin (400 µg mL^−1^); Azw + IAA BS, *A. brasilense* + exogenous auxin (30 µg mL^−1^); Azw + Azm, *A. brasilense* wild + *A. brasilense* mutant.

**Figure 8 microorganisms-10-02212-f008:**
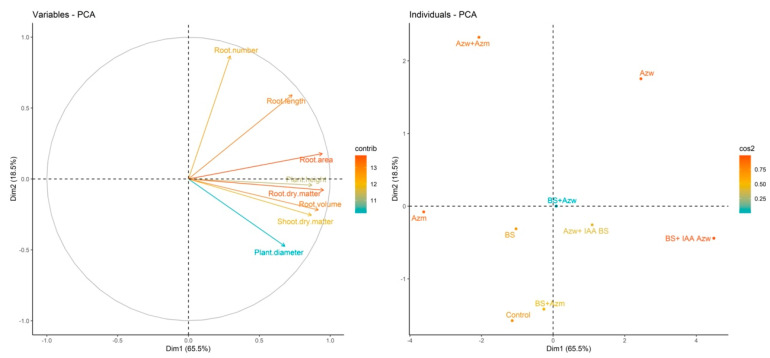
Main component analysis (PCA) for variables and treatments (individuals). Values on axes 1 and 2 represent the percentage of total variance explained by axes for the *dgt* genotype.

**Figure 9 microorganisms-10-02212-f009:**
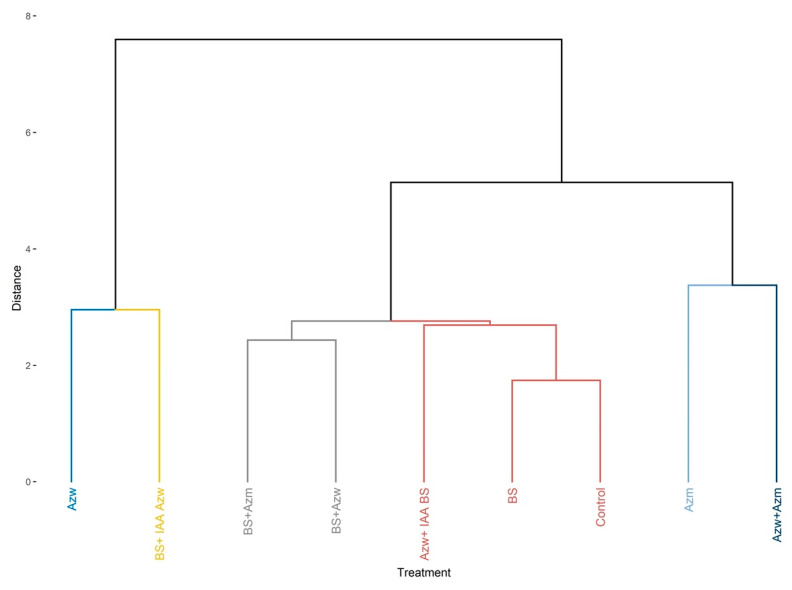
Hierarchical clustering of principal components (HCPCs) of inoculations tested.

**Table 1 microorganisms-10-02212-t001:** Description of treatments in the greenhouse with tomato genotypes Micro-Tom, *diageotropic* (*dgt*) and *Entire*.

Description of Treatments
Control (no inoculation)
*Bacillus subtilis* (BS)
*Azospirillum brasilense* wild (Azw)
*Azospirillum brasilense* mutant (Azm)
*B. subtilis* + *A. brasilense* wild (BS + Azw)
*B. subtilis* + *A. brasilense* mutant (BS + Azm)
*B. subtilis* + IAA Azw (BS + IAA Azw)
*A. brasilense* + IAA BS (Azw + IAA BS)
*A. brasilense wild* + *A. brasilense mutant* (Azw + Azm)

BS + IAA Azw means *B. subtilis* + 400 µg mL^−1^ exogenous auxin; AZW + IAA BS means *A. brasilense* + 30 µg mL^−1^ exogenous auxin.

**Table 2 microorganisms-10-02212-t002:** Effects of inoculation of the various treatments that differed from the control treatment in relation to the plant parameters analyzed in the tomato genotypes Micro-Tom, *Entire* and *dgt*.

Plant Parameter	Genotype	Effect	Treatments
Shoot Dry Mass	Micro-Tom	No differences	-
	*Entire*	Increased	Azw + IAA BS Azw + Azm
	*dgt*	No differences	-

Root Dry Mass	Micro-Tom	No differences	-
	*Entire*	Increased	Azw + IAA BS
	*dgt*	Increased	BS + IAA Azw

Plant Height	Micro-Tom	No differences	-
	*Entire*	No differences	-
	*dgt*	Reduced Increased	Azm BS + IAA Azw

Plant Diameter	Micro-Tom	Reduced	BS + Azw BS + Azm
	*Entire*	Increased	Azw BS + Azw Azw + IAA BS Azw + Azm
	*dgt*	Reduced	Azw + Azm

Root Length	Micro-Tom	Increased	Azm Azw + IAA BS
	*Entire*	Increased	BS + IAA Azw Azw + IAA BS Azw + Azm
	*dgt*	Increased	Azw BS + IAA Azw

Root Area	Micro-Tom	Increased	Azm
	*Entire*	Increased	BS + IAA Azw Azw + IAA BS Azw + Azm
	*dgt*	No differences	-

Root Volume	Micro-Tom	No differences	-
	*Entire*	Increased	Azw + IAA BS Azw + Azm
	*dgt*	No differences	-

Number of Roots	Micro-Tom	Reduced Increased	BS Azw + IAA BS
	*Entire*	Increased	Azw Azm BS + Azw BS + IAA Azw Azw + IAA BS Azw + Azm
	*dgt*	Increased	Azw Azw + Azm

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
