# Peer review of "Effect of Indole-3-Acetic Acid on Tomato Plant Growth"

_microorganisms, 2022, doi:10.3390/microorganisms10112212_

Round 1
Reviewer 1 Report
The research was interesting to me. However, it was disappointing that many of the manuscripts were not written correctly.
・The Article Type of the manuscript was not Review.
・The notation of indole-3-acetic acid in the Article Title was incorrect.
・In the manuscript, there were any sentences with a period in the wrong place.
・The correct abbreviation was not used (lines 107-108).
・Duplicate units (185 lines).
・References were formatted incorrectly (numbered by Referencess, not First author + year).
・Inability to fully understood Materials and Methods because they were not properly described.
There were many other mistakes, so please check the content thoroughly before considering submission.
Author Response
Reviewer: The research was interesting to me. However, it was disappointing that many of the manuscripts were not written correctly.
Answer: We authors thank the reviewer by the opportunity given to us for improving the manuscript.
Please see the changes done written in red color.
Reviewer:・The Article Type of the manuscript was not Review.
Answer; It has been changed
Reviewer:・The notation of indole-3-acetic acid in the Article Title was incorrect.
Answer: The title has been changed.
Reviewer:・In the manuscript, there were any sentences with a period in the wrong place.
Answer: It has been changed
Reviewer:・The correct abbreviation was not used (lines 107-108).
Reviewer:・Duplicate units (185 lines).
Reviewer:・References were formatted incorrectly (numbered by Referencess, not First author + year).
Answer: It has been changed.
Reviewer:・Inability to fully understood Materials and Methods because they were not properly described.
Answer: It has been changed. Could the reviewer to be more specific.
Reviewer:There were many other mistakes, so please check the content thoroughly before considering submission.
Answer: The manuscript has been reviewed.

Reviewer 2 Report
Overall the article is interesting and analyses the co-inoculation of two very important. PGPR. Comments are highlighted in the attached file.

Author Response
Reviewer Check for references, several of them are cited after the point. At the end, they must be numbered in order of appearance in the text (including table captions and figure legends) and listed individually at the end of the manuscript.
Answer: We authors want to thank the reviewer by the opportunity for improving our manuscript.
Answer: It has been checked.
Table1 There are many words in italic
Answer: The word coinoculation has been removed.
Answer: The words wild and mutant have been changed
Reviewer: mutants in which gene (s)?, please explain
Answer: The gene is auxin/IAA
Answer: The names of bacteria were written in italic.
Answer: The point at the end of software has been removed.
Answer: The double units have been removed µg IAA mL -1
Reviewer: I believe the table is not needed, the data is already in the text
Answer: Table 2 has been removed. The new table 2 was table 3.
Reviewer - Streptomyces fungus – It has been changed for bacteria
Reviewer: would change this sentence is wordy
Answer; It has been changed.

Round 2
Reviewer 1 Report
I found that my English improved greatly. However, there were many points where the explanation was insufficient to understand the research.
The three varieties of tomato used might have been selected due to their different sensitivities to auxin (indole-3-acetic acid; IAA), but the differences had not been sufficiently discussed.
In chapter 3, it was difficult to understand the order and explanation of the method. It seemed that 3.3 should be explained after 3.6. In 3.4, the centrifuged medium and supernatant were concentrated, but I thought that the centrifuged medium was the supernatant. In addition, under the condition of inoculating IAA instead of microorganisms, the concentration of IAA measured by concentrating was used, but I could not understand the basis for this. I thought that the cultured bacteria were inoculated together with the medium, but in that case, I thought that the concentration of IAA before concentration should be used. 3.4 and 3.5 seemed to be in reverse order. There was no description of the column used in HPLC for measuring the concentration of indole compounds. The last sentence of 3.5 seemed to relate not only to this section, but to all experimental results. I thought it should be explained in a separate section, including the statistical analysis. In 3.6, the concentration of bacteria to be inoculated to tomatoes seemed to be 10^8 cfu mL-1, but there was no explanation as to why this concentration was selected. In addition, although the seeds were soaked in the bacterial suspension, the volume at that time was not recorded. During the cultivation, 1 mL of bacterial suspension was inoculated every 15 days, but the concentration of the bacteria (especially the concentration of the mixture of two kinds of bacteria) was not stated. 3.7 seemed to be something that should be explained between 3.1 and 3.2. It seemed that 3.9 to 3.11 can be explained together with 3.8. In 3.10, it was not stated whether the alcohol used in preserving the roots was ethyl alcohol or other alcohol compounds.
At the beginning of Chapter 4, the concentrations of indole compounds were shown, but two types of Azw were listed and BS was not listed. Figures 1, 4, and 7 use Tukey's test as a statistical analysis method, but if you were using software, please explain in Chapter 3. If you were using software for principal component analysis, please explain in Chapter 3. Although the results of principal component analysis explain the correlation between treatment methods and plant parameters, such correlation couldn't be determined from the figures. If you were using software for a hierarchical component analysis, please explain in Chapter 3. You were doing a hierarchical component analysis and grouping, but I couldn't quite understand the need for that.
You concluded that the negative effect of bacterial mixture inoculation was not caused by excessive IAA, but I thought that it could be explained by experiments with only dgt genotype tomatoes (low auxin sensitivity). I thought that you should deepen the discussion in relation to the results of using the Entire genotype tomato, which was highly sensitive to auxin.